# First triple-wavelength lidar observations of depolarization and extinction-to-backscatter ratios of Saharan dust

Moritz Haarig[1], Albert Ansmann[1], Ronny Engelmann[1], Holger Baars[1], Carlos Toledano[2], Benjamin Torres[3], Dietrich Althausen[1], Martin Radenz[1], and Ulla Wandinger[1]

[1]Leibniz Institute for Tropospheric Research, Leipzig, Germany
[2]Atmospheric Optics Group, University of Valladolid, Valladolid, Spain
[3]Laboratoire d'Optique Atmosphérique, Université des Sciences et Technologies de Lille, Villeneuve d'Ascq, France

**Correspondence:** Moritz Haarig (haarig@tropos.de)

**Abstract.** Two layers of Saharan dust observed over Leipzig, Germany, in February and March 2021 were used to provide the first ever lidar measurements of the dust lidar ratio (extinction-to-backscatter ratio) and linear depolarization ratio at all three classical lidar wavelengths (355, 532 and 1064 nm). The pure dust conditions during the first event exhibit lidar ratios of 47±8, 50±5 and 69±14 sr and particle linear depolarization ratios of 0.242±0.024, 0.299±0.018 and 0.206±0.010 at the wavelengths of 355, 532 and 1064 nm, respectively. The second, slightly polluted dust case shows a similar spectral behavior of the lidar and depolarization ratio with values of the lidar ratio of 49±4, 46±5 and 57±9 sr and the depolarization ratio of 0.174±0.041, 0.298±0.016 and 0.242±0.007 at 355, 532 and 1064 nm, respectively. The results were compared with Aerosol Robotic Network (AERONET) version 3 inversion solutions and the Generalized Retrieval of Aerosol and Surface Properties (GRASP) at six and seven wavelengths. Both retrieval schemes make use of a spheroid shape model for mineral dust. The spectral slope of the lidar ratio from 532 to 1064 nm could be well reproduced by the AERONET and GRASP retrieval schemes. Higher lidar ratios in the UV were retrieved by AERONET and GRASP. The enhancement was probably caused by the influence of fine-mode pollution particles in the boundary layer which are included in the columnar photometer measurements. Significant differences between the measured and retrieved wavelength dependence of the particle linear depolarization ratio were found. The potential sources for these uncertainties are discussed.

## 1 Introduction

Triple-wavelength polarization Raman lidar observations of particle depolarization and extinction-to-backscatter ratios are of importance for several reasons. Many lidars (including ceilometers) are standard backscatter lidars and need trustworthy information about the particle extinction-to-backscatter ratio (lidar ratio) in the retrieval of backscatter and extinction profiles and aerosol optical depth (AOD). This is of special importance in the case of the spaceborne CALIPSO lidar (Cloud-Aerosol Lidar and Infrared Pathfinder Satellite Observations) monitoring aerosol and clouds around the globe at 532 and 1064 nm (Omar et al., 2009; Kim et al., 2018). Besides CALIPSO, the CATS space lidar (Cloud Aerosol Transport System, Yorks et al., 2016) will benefit from lidar ratio measurements at 1064 nm, as well as possible future space missions operating a backscatter lidar at 1064 nm.

Triple-wavelength lidar measurements are also important for aerosol typing efforts that make use of spectrally resolved intensive aerosol parameters such as the linear depolarization ratio, lidar ratio, as well as the extinction and backscatter-related Ångström exponents for all climate-relevant aerosol types like marine aerosol, dust, smoke, volcanic ash, and haze particles (Burton et al., 2012; Groß et al., 2013; Baars et al., 2017). In this respect, our measurements will contribute to these aerosol typing efforts by adding new information on dust lidar and depolarization ratios at 1064 nm.

The most important aspect of triple-wavelength lidar observations of dust depolarization and lidar ratios is, however, that such measurements at all three classical aerosol lidar wavelengths (355, 532, and 1064 nm) are required to improve optical models applied to simulate the optical properties of aerosol particles as a function of size distribution, shape characteristics, and chemical composition. Especially in the case of mineral dust there is a strong request for those lidar observations to improve the applied particle shape parameterization. The optical properties of dust strongly depend on particle shape (depolarization ratio), size and mineralogy (lidar ratio). These data, presented here, will support the validation of shape models for the irregularly shaped dust particles.

Modeling of dust optical properties is especially challenging for the 180° backscattering direction. Complex dust shape models were developed to improve the agreement between lidar observations and respective simulation results (Gasteiger et al., 2011; Kemppinen et al., 2015; Saito et al., 2021). The spheroidal shape model is widely used (Dubovik et al., 2006), e.g., in the analysis of Aerosol Robotic Network (AERONET, Holben et al., 1998) sun and sky radiometer measurements to obtain the inversion products which include the lidar ratio and the particle linear depolarization ratio at four wavelengths between 440 and 1020 nm. The photometer retrieval products were compared with lidar measurements of desert dust (Tesche et al., 2009; Müller et al., 2010, 2012; Noh et al., 2017; Shin et al., 2018; Toledano et al., 2019) and discrepancies were found.

Triple-wavelength lidar observations of the depolarization ratio are available since 2013 (Burton et al., 2015; Haarig et al., 2017). However, direct observations of the lidar ratio at all three lidar wavelengths are missing as was stated in Shin et al. (2018). Haarig et al. (2016) showed, for the first time, directly measured vertical profiles of the 1064 nm extinction coefficient. Since then, we used the triple-wavelength Raman lidar approach to characterize cirrus (Haarig et al., 2016) and wildfire smoke (Haarig et al., 2018) in terms of backscatter, extinction, lidar ratio, and depolarization ratio at 355, 532 and 1064 nm. Now, we present two case studies of pure dust and polluted dust outbreaks towards central Europe which occurred in February and March 2021. The first Saharan dust plume extended from the ground up to 8 km height and reached our station in Leipzig, Germany, in less than 2 days (around 36 hours) after emission. The fast transport process at great heights prevented mixing with anthropogenic pollution. The second dust outbreak occurred one week later. The dust spent more time over Europe (3–4 days of transport) and the dust plume mixed with European haze.

The article is structured as follows. After a short description of the lidar system and the additional, new option to measure extinction coefficients at 1064 nm in Sect. 2, the two observed dust cases are presented in Sect. 3. The discussion in Sect. 4 focuses on the Ångström exponents and the spectral dependence of the depolarization ratio and lidar ratio. We use the opportunity to compare our results with AERONET v3 inversion results as well as with products obtained by applying the Generalized Retrieval of Aerosol and Surface Properties (GRASP) technique (Dubovik et al., 2014; Torres et al., 2017; Dubovik et al., 2021), available for the same dust event in February 2021.

## 2 Instrumentation

For the present study two lidars and an AEROENT sun photometer at the Leibniz Institute for Tropospheric Research (TROPOS) in Leipzig (51.35°N , 12.43°E), Germany, were used.

The multiwavelength polarization Raman lidar BERTHA (Backscatter Extinction lidar-Ratio Temperature Humidity profiling Apparatus, Haarig et al., 2017) provides the backscatter coefficient, extinction coefficient and depolarization ratio at 355, 532 and 1064 nm (3+3+3 configuration). However, it can not measure the depolarization ratio and the extinction coefficient at 1064 nm at the same time. A couple of minutes are necessary to switch the NIR setup from depolarization to extinction measurements.

The radiometric or Rayleigh calibration of the signals was done in clear air which is at 1064 nm a challenging task. At 1064 nm the elastic return signal from molecules is 81 times and 16 times smaller than for 355 and 532 nm, respectively. The reference height for the 1064 nm signal was set some hundreds of meters above the dust layer top height for a sufficient strong molecular signal. On 3 March 2021 (Sect. 3.2) a thick cirrus cloud was present, which could be used to check the Rayleigh calibration. A backscatter color ratio (1064/532) of 0.88 was observed, which is in line with observations by Vaughan et al. (2010) who reported values of 1.01±0.25 in cirrus clouds. On 22 February 2021 (Sect. 3.1), the cirrus clouds were too thin to check the calibration.

The extinction coefficients at 355 and 532 nm were derived by using the vibrational-rotational Raman channels at 387 and 607 nm, respectively. The rotational Raman technique (Whiteman, 2003a, b; Veselovskii et al., 2015; Haarig et al., 2016; Ortiz-Amezcua et al., 2020) was applied to provide measurements of the extinction coefficient at 1064 nm. A detailed description of rotational Raman technique implemented in BERTHA can be found in Haarig et al. (2016). The Raman channel is equipped with an interference filter centered at 1058 nm (Alluxa, Santa Rosa, CA, http://www.alluxa.com). The transmission band is from 1053 to 1062 nm with a transmission >90% in this wavelength range. The filter bandwidth of 9 nm restricts the 1058 nm Raman observations to nighttime hours. For the laser wavelength of 1064.14 nm, the transmission is specified to be 0.005%. For a better suppression of the transmitted radiation at 1064 nm two identical 1058 nm interference filters are used in front of the photomultiplier tube (photocounting, R3236 from Hamamatsu, Japan). However, still some cross-talk from the elastic signal at 1064 nm was present in the Raman channel. The enhanced signal received from the liquid clouds on 22 February 2021 (Section 3.1) was used to estimate the cross-talk factor of 0.0175±0.0005. For the further analysis, the elastic signal at 1064 nm multiplied by the cross-talk factor was subtracted from the Raman signal at 1058 nm to correct for the cross talk. In this simple approach, the cross talk is assumed to be temperature independent. The temperature dependent transmission of the oxygen and nitrogen Raman lines varies from 0.729 to 0.793 between 288 and 208 K, which was the maximum and minimum temperature observed in the troposphere on 22 February 2021. The rotational Raman signal at 1058 nm was corrected for the temperature dependence after the cross-talk correction. Within the dust layer, the correction led to around 3% higher extinction values. Detailed calculations of the temperature dependence of rotational Raman lidar signals can be found in Veselovskii et al. (2015).

In BERTHA, linearly polarized laser pulses are transmitted into the atmosphere and the so-called co-polarized and cross-

polarized signal components are measured. "Co" and "cross" denote the planes of polarization parallel and orthogonal to the plane of linear polarization of the transmitted laser pulses, respectively. In the case of spherical droplets, wet marine and small haze particles, the polarization state of the laser pulse radiation is preserved during the backscattering process and the particle linear depolarization ratio (PLDR), computed from the ratio of the cross-polarized to the co-polarized signal component, is close to zero. In the case of irregularly shaped dust particles, depolarization takes place during the backscattering event and the PLDR values are close to 0.3 at 532 nm (Freudenthaler et al., 2009; Haarig et al., 2017). The calibration of the polarization-sensitive channels at all three wavelengths makes use of the so-called $\Delta 90°$ method (Freudenthaler, 2016). The error estimation for the BERTHA system is described in Haarig et al. (2017).

The second lidar was the Polly[XT] (Engelmann et al., 2016), . It is described below. The second lidar used in the present study is a Polly[XT] (POrtabLe Lidar sYstem, neXT generation, Engelmann et al., 2016) which is usually used for ship-borne observations (Engelmann et al., 2021) within OCEANET Atmosphere. It is a Raman polarization lidar, measuring the backscatter coefficient at 355, 532 and 1064 nm, the extinction coefficient at 355 and 532 nm (vibrational-rotational Raman) and the depolarization ratio at 355 and 532 nm. Additionally, near range channels at 355, 387, 532 and 607 nm provide information at lower altitudes and the dual-field-of-view polarization technique (Jimenez et al., 2020) is implemented to study liquid clouds. The Polly[XT] was operated several meters apart from the BERTHA instrument.

A CIMEL sun-sky photometer of AEROENT (Holben et al., 1998) is operated at Leipzig since 2000. Currently, only level 1.5 data are available for February and March 2021, although the data already passed all quality criteria, including cloud screening and operational checks (Giles et al., 2019). Besides the standard AERONET data analysis procedure (Dubovik and King, 2000; Dubovik et al., 2006; Sinyuk et al., 2020), the related GRASP retrieval scheme was applied (Dubovik et al., 2014; Torres et al., 2017; Toledano et al., 2019; Dubovik et al., 2021). GRASP is based on a similar approach as the standard AERONET retrieval (Dubovik and King, 2000), but using optical information (AOD and radiances) at the wavelengths of 380, 500 and 1640 nm additionally. From both algorithms the column lidar ratios and depolarization ratios at several wavelengths were retrieved.

## 3 Observations

Two case studies of triple-wavelength depolarization ratio and lidar ratio observations of Saharan dust from late winter 2021 will be shown. Pure dust conditions were observed on 22–23 February 2021, while mixed pollution-dust conditions prevailed on 3 March 2021.

### 3.1 First dust outbreak: 22–23 February 2021

Huge amounts of mineral dust were emitted from western Saharan regions around and before 21 February 2021. The dust plumes were directly transported towards central Europe and reached Leipzig in less than 2 days (around 36 hours) after emission as shown by the HYSPLIT (HYbrid Single-Particle Lagrangian Integrated Trajectory, Stein et al., 2015) backward trajectories in Figure 1b. More detailed source attributions indicate the Sahara as main source region for the lofted layers (see Fig. 2). Below 3 km height, aerosol from continental Europe were probably mixed into the Saharan dust plumes. In the night

of 22–23 February 2021, the dust plumes reached from the ground up to 8 km height. Saharan dust was transported towards central Europe continuously from the evening of 22 February till the morning of 26 February 2021. The whole dust event was studied using the continuous observation of the Polly$^{XT}$ lidar (polly.tropos.de/calendar/location/1, Baars et al., 2016). The transport pattern were similar to the dust outbreak in October 2001 described in Ansmann et al. (2003), which was measured by the entire EARLINET community (European Aerosol Research LIdar NETwork, now part of the Aerosol, Clouds and Trace Gases Research Infrastructure, ACTRIS, Pappalardo et al., 2014).

The lidar measurements are presented in Fig. 1. The time-height plot (Fig. 1a) of the cross-polarized and range-corrected signal at 532 nm provides an overview of the dust conditions observed with the BERTHA lidar system. During the first 20 min the system was in the configuration to measure the depolarization ratio at 1064 nm. Then, the configuration was switched to permit measurements of the extinction coefficient at 1064 nm for the next 2 h 15 min. Some clouds were forming in the humid dust layer at around 4 km height, indicated by strong backscatter signals (dark red) and the attenuation of laser radiation above the cloud layers. The cloud-containing profiles were excluded from the calculation of the dust optical properties, but used for the estimation of the cross-talk correction. The signals at 355 nm of the BERTHA lidar were very weak during that night. In fact, neutral density filters in front of the UV channels were not adapted to the strongly reduced power of the UV laser. Therefore, the optical properties at 355 nm were taken from the nearby Polly$^{XT}$ measurements. Additionally, the optical properties at 532 nm measured with the Polly$^{XT}$ are shown in Fig. 1c–f. The profiles at 532 nm (green lines) from the different lidar observations agree well.

In the dust layer (main part 3–5 km height, extending up to 8 km height), no wavelength dependence for the backscatter coefficient between 355 and 532 nm was found (backscatter Ångström exponent BAE = 0.040±0.524). The backscatter coefficient at 1064 nm was around 80% of the value at 532 nm within the dust layer. The extinction coefficient exceeded 100 Mm$^{-1}$ at around 3.5 km height at all three wavelengths. The total AOD was approximately 0.45 at 532 nm during that night. The extinction Ångström exponent (EAE, Fig. 1e) were vertically constant within the dust layer and close to zero, but slightly negative. The lidar ratio (Fig. 1f) derived in the strongest part of the dust layer (3–5 km height), increased with wavelength from 47±8 sr at 355 nm, to 50±5 sr at 532 nm and 69±14 sr at 1064 nm. The particle linear depolarization ratio PLDR at 532 nm of up to 0.30±0.02 (3–5 km layer mean and systematic uncertainty) indicates pure dust conditions (Freudenthaler et al., 2009). The PLDR decreases towards 355 nm (0.242±0.024) and 1064 nm (0.206±0.010). In order to compare simultaneous observations of the PLDR the values are reported for the first 20 min (22:12–22:32 UTC) when the 1064 nm channels measured the depolarization ratio. Whereas all other intensive optical properties were measured from 22:45–01:02 UTC. The depolarization ratios for the later period were nearly the same (0.256±0.025 and 0.298±0.017 at 355 and 532 nm, respectively). A summary of the intensive optical properties measured in the night of 22–23 February 2021 can be found in Table 1.

The AERONET version 3 inversion results for the lidar ratio and PLDR (five separate observations) performed on 23 February 2021 between 11:35 and 14:49 UTC will be compared with the lidar observations in Section 4. The AERONET measurement period is indicated in Fig. 2. The continuous observations of the Polly$^{XT}$ and the source attribution for 23 February 2021 in Fig. 2 reveal that the same Saharan dust layer was present during the nighttime lidar measurements and the daytime AERONET observations. At lower heights, Europe is indicated besides the Sahara as a potential source region. The AERONET measure-

160 ments from 22 February 2021 did not capture the Saharan dust event, which arrived at 19 UTC at Leipzig, first slices already at 16 UTC.

## 3.2 Second dust outbreak: 3 March 2021

A second dust outbreak occurred one week later on 3 March 2021. Travel time was 3–4 days from the Sahara to Leipzig via Spain and France as indicated by the HYSPLIT backward trajectories in Fig 3b. On 3 March 2021, the dust plume arrived
around 10 UTC over Leipzig with some cirrus clouds above. In the night, the dust plume reached its largest vertical extend. Some clouds were formed within the dust layer. In the morning of 4 March, around 6 UTC, precipitation started and washed out the dust. The development of the dust plume over Leipzig was studied using the continuous observation of PollyNET (https://polly.tropos.de/calendar/location/1, Baars et al., 2016).

This time the BERTHA lidar was fully operational at all three wavelengths so that also the optical properties at 355 nm could
be used. The lidar observations are shown in Fig. 3. The dust layer extended up to 5 km in the beginning, later on the dust plume descended down to 4 km height. During the first 20 min, the lidar was configured to measure the depolarization ratio at 1064 nm (3+2+3 configuration). The process of changing the interference and polarization filters was optimized so that 7 min later the extinction measurement at 1064 nm could be started (3+3+2 configuration). To avoid too strong vertical smoothing for the extinction coefficient at 1064 nm compared to the previous case, signal profiles collected over 3 h 20 min (21:11–00:32
UTC) were averaged. The main dust layer extended from 1.5–4 km height and provided vertically homogeneous conditions for averaging. The profiles of the extinction coefficients and lidar ratios in Fig. 3d and f are shown for a sliding average window length of 750 m.

The AOD was 0.41 at 355 nm, 0.37 at 532 nm and 0.34 at 1064 nm. The lidar ratio in the center of the dust layer (2.0–3.5 km height) at 1064 nm (57±9 sr) was slightly higher than at 532 nm (46±5 sr). The slight increase in the lidar ratio at 355 nm
(49±4 sr) compared to 532 nm could be an indication for pollution mixed into the dust layer.

The depolarization ratios (Fig. 3g) were measured from 20:44–21:04 UTC. The dashed lines indicate the depolarization ratios at 355 and 532 nm (layer mean values (2.0–3.5 km height) of 0.179±0.053 and 0.264±0.0163, respectively) measured at the same time as the other optical properties (21:11–00:32 UTC). The dust layer showed a larger vertical extent up to 5 km height and higher values of the PLDR at 532 nm (0.292±0.016) during the first shorter measurement period. Especially, the PLDR at
185 355 nm was significantly reduced (0.174±0.041) compared to the previous case with pure dust conditions. The longer transport over Europe obviously caused the weak mixing with anthropogenic pollution. A summary of the intensive optical properties measured on 3 March 2021 can be found in Table 1. Unfortunately, no AERONET observations could be used for comparison. The dust layer was accompanied by extended cirrus layers preventing sun photometer observations and on the next day the precipitation event removed the dust.

## 4 Discussion

### 4.1 Backscatter and extinction Ångström exponent

The aerosol optical depth from the lidar measurement in the night of 22–23 February 2021 was 0.48±0.03 at 355 nm, 0.45±0.03 at 532 nm and 0.45±0.03 at 1064 nm. Below the dust layer, the backscatter coefficient times a lidar ratio of 50 sr was used to estimate the extinction coefficient. The uncertainty was estimated by varying the lidar ratio ±10 sr. Figure 2 shows a slight decrease in the dust plume height till the next morning where the AERONET observations were performed. Slightly lower AODs of 0.43 at 340 nm, 0.42 at 500 nm and 0.36 at 1020 nm were measured by the photometer (AERONET, 2021).

The intensive optical properties such as the extinction Ångström exponent (EAE) are given in Table 1. The lidar derived EAE for the three wavelength are all negative, but close to zero ranging between –0.005 (355/532) and –0.098 (355/1064). Negative EAE (355/532) down to –0.2 were observed for Saharan dust in Senegal (Veselovskii et al., 2016), where the EAE of Saharan dust accumulated in the interval –0.2 to 0.2.

The AERONET observations from the early morning of 23 February 2021 showed an Ångström exponent calculated from the AOD at 440 and 870 nm of 0.217±0.039 and for the wavelengths 380/500 of 0.124±0.073 (AERONET, 2021). The greater and positive Ångström exponent of the AERONET observations might be influenced by the small urban haze particles in the wintertime planetary boundary layer. The lidar data did not include these urban haze effects. Considering the whole column, AOD Ångström exponents from the lidar observations of 0.155±0.177 (355/532), 0.008±0.021 (532/1064) and 0.062±0.015 (355/1064) were obtained. The AOD Ångström exponent in the UV–VIS range agrees within the uncertainties, whereas the higher AOD at 1064 nm led to lower Ångström exponents from the lidar, probably due to an overestimation of the AOD at 1064 nm in the boundary layer.

The backscatter Ångström exponents (BAE) are provided in Table 1. The backscatter coefficients at 355 and 532 nm were almost the same (100±5%), whereas values at 1064 nm were lower (here 79% of the value at 532 nm) as found in other observations of mineral dust (Veselovskii et al., 2016; Hofer et al., 2020). This behavior is expressed in the BAE (532/1064) of 0.35, and at Barbados of 0.47. Liu et al. (2008) reported CALIPSO observations of the backscatter color ratios (1064/532) of 0.72–0.75 in Saharan dust. This translates to an average BAE of 0.44, which is comparable to the value observed in the Saharan air layer at Barbados.

### 4.2 Spectral dependence of the lidar ratio

The discussion of the spectral behavior of the lidar ratio of Saharan dust is based on the two presented case studies from Leipzig and observations at Barbados during the Saharan Aerosol Long-range Transport and Aerosol–Cloud-Interaction Experiment (SALTRACE, Weinzierl et al., 2017). The BERTHA lidar system at Barbados (13°N, 59°W) measured the Saharan dust after long-range transport over 5000–8000 km across the Atlantic Ocean in three intensive campaigns in 2013 and 2014 (Haarig et al., 2017). The statistics of the lidar ratio at 355 and 532 nm and the Ångström exponents based on 22 SALTRACE measurement cases are reported in the present publication for the first time (Table 1). During the SALTRACE campaigns, the rotational Raman technique applied to measure the extinction at 1064 nm was not implemented in the BERTHA lidar. However,

the lidar ratio at 1064 nm could be estimated for one intense dust event with very homogeneous dust conditions observed on 20 June 2014. Here, the column-integrated backscatter coefficient at 1064 nm in the dust layer after sunset and the AOD at 1020 nm (from AERONET observations) before sunset were used as described in Mamouri and Ansmann (2017). An AOD contribution of 0.035 for marine aerosol below the dust layer was subtracted to obtain the pure dust value for the lidar ratio estimate. This method led to a dust lidar ratio of 67$\pm$15 sr at 1064 nm.

The spectral dependence of the lidar ratio of Saharan dust is shown in Fig. 4a. The lidar ratio does not change significantly in the wavelength range of 355 to 532 nm. In the case of Saharan dust, this behavior was observed in numerous studies (e.g., Tesche et al., 2011; Groß et al., 2015). Some studies however, point to higher values in the UV (Mattis et al., 2002). Veselovskii et al. (2020) observed cases with the same lidar ratio at 355 and 532 nm and cases with higher lidar ratios at 355 nm. The ratio of lidar ratios for 355 and 532 nm is an indicator of the imaginary refractive index enhancement in the UV depending on the mineralogical composition of the dust particles (Veselovskii et al., 2020).

The present study indicates an increase of the lidar ratio from 532 to 1064 nm. As mentioned, the reason is the lower particle backscatter coefficient $\beta$ at 1064 nm compared to the one at 532 nm so that the backscatter Ångström exponent $BAE$, defined as $BAE_{\lambda_i,\lambda_j} = \ln(\beta_i/\beta_j)/\ln(\lambda_j/\lambda_i)$ in the spectral range from wavelength $\lambda_i$ to $\lambda_j$, is positive. The extinction coefficient $\alpha$ usually does not show a significant wavelength dependence so that the respective extinction Ångström exponent $EAE$ is close to zero as shown in Fig. 1e and 3e. According to Ansmann et al. (2002), the Ångström exponent $SAE$ for the lidar ratio $S$, given by

$$SAE = EAE - BAE, \tag{1}$$

is then negative. This relationship (1) holds within the uncertainties for the intensive optical properties provided in Table 1.

The ratios of the lidar ratios given in Table 1 quantify the increase of the lidar ratio from 532 to 1064 nm. We found around 24–38% higher values for the lidar ratio at 1064 nm compared to 532 nm. A similar increase (27–34%) in lidar ratio from 532 to 1064 nm was observed from three early CALIPSO measurements of Saharan dust transported from Africa across the Atlantic (Liu et al., 2008). Combined sun photometer and lidar observations during the Saharan Mineral Dust Experiments (SAMUM) in Morocco led to the conclusion that the lidar ratio is almost independent of wavelength with values of 55$\pm$5 sr, 56$\pm$5 sr, and 59$\pm$7 sr at 355, 532, 1064 nm, respectively (Tesche et al., 2009). The authors used the extinction Ångström exponent (for the 500-1020 nm wavelength range) from sun photometer observations to transfer the extinction profile from 532 to 1064 nm and calculate the lidar ratio at 1064 nm using the corresponding backscatter profile in addition. In contrast to those measurements for freshly emitted dust close to its source, we found now lidar ratio values of 47$\pm$8 sr, 50$\pm$5 sr, and 69$\pm$14 sr for the three wavelengths after a dust transport of less than 2 days.

The spectral dependence of the lidar ratio measured with lidar and retrieved from the AERONET observations is compared in Fig. 4b. The AERONET data base (AERONET, 2021) provides AOD values at 440, 675, 870 and 1020 nm together with further products such as lidar ratio and linear depolarization ratio. The lidar ratio and particle linear depolarization ratio are retrieved by means of the AERONET v3 inversion algorithm (Dubovik and King, 2000; Giles et al., 2019; Sinyuk et al., 2020) which assumes a spheroidal dust particle shape (Dubovik et al., 2006). Additionally to the standard four wavelengths, the inversion

was performed using the GRASP algorithm at six wavelengths (plus 500 and 1640 nm) and seven wavelengths (plus 380 nm, Dubovik et al., 2014; Torres et al., 2017; Dubovik et al., 2021). For the mean values the data were filtered in the way that the residual is <10% of the inversion retrieval. Since radiance measurements were included in the GRASP data analysis the refractive index and the sphericity are not pre-set and are retrieved following the same strategy as AERONET but with the extra information provided by the use of more wavelengths. A comparison of the lidar observations with the respective AERONET v3 inversion products was only possible on 22–23 February 2021.

The spectral slope of the lidar ratio is well reproduced by the AERONET observations for the spectral range from 675 to 1020 nm. The seven-wavelength mean values show exactly the same spectral slope as obtained by the lidar observations. The retrieved lidar ratios at 440 nm using the standard AERONET algorithm indicate higher values of the lidar ratio in the UV range. The GRASP retrieval at 440 nm shows a much better agreement with the lidar observations. The increase of the lidar ratio at 380 nm (seven-wavelength retrieval) points again to higher values of the lidar ratio in the UV range. The enhanced lidar ratios in the UV retrieved by AERONET were already discussed by Shin et al. (2018): The increase of the imaginary part of the refractive index at 440 nm compared to 675 nm is too strong in the AERONET inversion procedure, resulting in too high lidar ratios at 440 nm. However, in the present case the explanation might be simpler. The sun photometer measures columnar values whereas the profile measurement of lidar permits to focus on the dust layer only. The fine-mode aerosol pollution of the wintertime boundary layer at Leipzig can significantly influence the AERONET lidar-ratio products, especially at the shorter wavelengths of 440 and 380 nm. Therefore, the columnar lidar ratio should increase towards the UV.

As mentioned, the refractive index depends on the mineralogical composition of dust particles and this changes for different dust source regions. Schuster et al. (2012) investigated the variation of the lidar ratio obtained from AERONET observations for different deserts around the Earth and found significant differences. The highest lidar ratios were found for the western Sahara, lower values for Asian deserts. The global coverage of AERONET sun photometers is a big advantage. However, the retrievals should be checked against Raman or high spectral resolution lidar (HSRL) observations as has been started by Shin et al. (2018), ideally in a desert, where no other aerosol type contributes to the columnar values. The present study indicates a good agreement of the lidar ratio in the spectral range of 675 to 1020 nm.

## 4.3 Spectral dependence of the depolarization ratio

The spectral dependence of the particle linear depolarization ratio of Saharan dust is shown in Fig. 5a. The results of the two presented case studies are set into context with previous triple-wavelength depolarization ratio observations. The observations of Saharan dust close to its source in Morocco (Freudenthaler et al., 2009) and after long-range transport towards Barbados (Haarig et al., 2017) and North America (Burton et al., 2015) show a consistent spectral slope. The PLDR increases in the wavelength range from 355 to 532 nm and then decreases again towards the wavelength of 1064 nm.

Table 1 provides the values for the two presented cases studies at Leipzig and the SALTRACE observations. The table includes the ratio of PLDR for the wavelength pairs of 355 and 532 nm and of 1064 and 532 nm. The depolarization ratio at 355 nm was very low on 3 March 2021 (0.174±0.041) and probably reflects the impact of aerosol pollution mixed into the dust layers. The small and spherical pollution particles affect the backscattering at shorter wavelengths more effectively than at the longer

wavelengths. For pure Saharan dust (22–23 February 2021, Barbados, Morocco), the PLDR at 355 nm reached around 80–90% of the value at 532 nm. The PLDR at 1064 nm depends on the amount of rather large dust particles in the observed layer and thus is very sensitive to the travel duration (and the corresponding removal of large particles). The decrease of the PLDR with wavelength in the range from 532 to 1064 nm is expressed by PLDR ratios of 0.69–0.83 for Saharan dust after long-range transport (Table 1).

The spectral slope of the PLDR is not an unique feature of Saharan dust. Hu et al. (2020) observed the same spectral dependence but with generally higher values for dust from the Taklamakan desert in western China. Observations were performed very close to the main dust sources, and thus with an enhanced fraction of very large dust particles. The same behavior was found for locally emitted dust in the Southwest of the United States (Burton et al., 2015). The large dust particles of the freshly emitted dust were still present in the air during the lidar observations and produced significantly higher depolarization ratios of $0.38\pm0.01$ at 1064 nm.

The spectral dependence of the PLDR in Fig. 5b derived from the sun and sky photometer observations is very different from the respective lidar observations of the PLDR spectral behavior. The lidar measurements of desert dust generally show a pronounced maximum of the depolarization ratio in the visible wavelength range. This is not visible in the AERONET spectra. By using the GRASP method (including the use of optical data measured at 1640 nm, not shown in the figure) leads to a decrease of the PLDR with wavelength, starting already at 1020 nm. However, the PLDR in the NIR is still overestimated for the presented case, in which a PLDR of $0.206\pm0.010$ at 1064 nm was measured. Compared to previous observations of the PLDR at 1064 nm (Fig. 5), the value is quite low. The six- and seven-wavelength GRASP retrievals lead to a better agreement in the NIR, but underestimates the depolarization ratio at the shorter wavelengths. Previous comparisons of lidar-measured and AERONET-retrieved depolarization ratios pointed out that the spheroidal model used for the AERONET inversions fails to predict the spectral slope of the PLDR (Noh et al., 2017; Shin et al., 2018). Only Toledano et al. (2019) was able to obtain the spectral slope as observed with polarization lidar systems by using the six-wavelength sun photometer retrieval for Saharan dust observations on Barbados during SALTRACE. In their study, the PLDR increases from $0.277\pm0.040$ at 440 nm to $0.282\pm0.031$ at 675 nm and then decreases again to $0.259\pm0.030$ at 1020 nm. A further decrease was retrieved for the wavelength of 1640 nm $(0.191\pm0.028)$. Again, the boundary-layer fine-mode pollution (over Leipzig) influenced the AERONET retrievals and may lead to smaller AERONET depolarization ratio values than observed with lidar in the lofted dust layer. However, the significant discrepancies between the AERONET and the lidar observations in the 532–1064 nm spectral range remain an unsolved issue and seem to be related to the used spheroidal shape model in the AERONET and GRASP retrieval schemes.

## 5 Conclusions

We presented, for the first time, measurements of the dust lidar ratio at the main aerosol lidar wavelengths of 355, 532 and 1064 nm. Together with the depolarization ratio at these wavelengths, we demonstrated the unique potential of a triple-wavelength polarization Raman lidar to permit so-called 3+3+3 profiling (3 backscatter coefficients, 3 extinction coefficients, 3 depolarization ratios). Two case studies of Saharan dust over central Europe were presented, one for pure dust conditions and

another case with slightly polluted dust. The measurement of the depolarization ratio and lidar ratio at three wavelengths from 355 to 1064 nm provides an improved, more complete set of constraints in modeling efforts to develop a realistic dust particle shape model. A better assessment of the absorption properties which are linked to the mineralogical composition of the dust particles may be possible by considering the spectral information of the lidar ratio at 355, 532 and 1064 nm. Nevertheless, realistic assumptions on the imaginary part of the refractive index are required. Good measurements of the refractive index are important to allow for an accurate computation of the spectral slope of the lidar ratio in the wavelength range from 355 to 1064 nm.

To derive the extinction coefficient at 1064 nm from backscatter lidars and ceilometers, realistic estimates of the lidar ratio at this wavelength are necessary. We presented now the first direct measurements of the lidar ratio at 1064 nm for Saharan dust. It would be desirable, in this context, to perform such triple-wavelength lidar measurements in very different desert regions of the world. Middle East and central Asian dust show, e.g., lower lidar ratios, especially at 532 nm, as presented here (Hofer et al., 2020). We found lidar ratios of 47±8, 50±5 and 69±14 sr at 355, 532 and 1064 nm, respectively, for pure dust conditions, and respective lidar ratios of 49±4, 46±5 and 57±9 sr for slightly polluted dust. In central Asia, pure dust lidar ratios were on average 43±3 and 39±4 sr at 355 and 532 nm, respectively (Hofer et al., 2020).

The lidar ratio at 532 and 1064 nm are important input parameters for the CALIPSO lidar data analysis to derive the respective extinction coefficients at these two wavelengths (Omar et al., 2009). In the retrieval algorithm version 4, lidar ratios of 44±9 sr at 532 nm and 44±13 sr at 1064 nm are used for the aerosol type of pure dust (Kim et al., 2018). Based on our results, we would suggest to use a higher value at 1064 nm compared to 532 nm.

The comparison with AERONET v3 inversion products revealed that the spectral dependence of PLDR was not matched by the AERONET results. The GRASP computations based on the optical information (AOD, radiances) measured at six and seven wavelengths improved the agreement, but lead to overall lower values. In contrast to lidar systems which are only operating at 180° backscatter angle, this angle is not accessible for the photometer. Therefore, the photometer retrievals are not optimized for this special angle. Joint inversions of the lidar and photometer data as done in Lopatin et al. (2021) would be desirable.

The lidar measurements could confirm the spectral slope of the lidar ratio from the VIS towards the NIR retrieved by the AERONET and GRASP data analysis. The fine-mode pollution aerosol in the boundary layer led to higher lidar ratios in the UV from the columnar photometer measurements compared to the Saharan dust observations with lidar.

We may conclude that an appropriate particle shape model for the non-spherical dust particles is still missing. The spheroidal shape model (Dubovik et al., 2006) has proven to be useful to derive volume and surface area concentrations from sky radiance and AOD observations at several wavelengths even for non-spherical dust particles. However, this shape model is not able to reproduce the spectral slope of the particle linear depolarization ratio with a maximum at 532 nm measured with lidar at the 180° backscatter direction. Complex particle-shape models (Gasteiger et al., 2011; Kemppinen et al., 2015; Saito et al., 2021) are computationally expensive. Nevertheless, they may lead to a more realistic representation of the non-spherical dust particles for the 180° backscatter direction in the models.

As a final remark, it would be helpful, and a good addition to field observations, if laboratory measurements of the depolarization and lidar ratios at all three wavelengths (in the 180° backscatter direction) could be realized for well defined size fractions

of real dust particles with real irregular shape characteristics. An effort was already started by Miffre et al. (2016).

*Data availability.* The BERTHA lidar data can be obtained upon request from Moritz Haarig (haarig@tropos.de). The Polly[XT] data are available in the PollyNET data base (https://polly.tropos.de/) (PollyNET, 2021). The AERONET data are available via the AERONET web page (https://aeronet.gsfc.nasa.gov/) under the site 'Leipzig' (AERONET, 2021).

*Author contributions.* MH performed the lidar measurements, analyzed the data and wrote the manuscript. AA provided advise and valuable
comments on the manuscript. RE supported the rotational Raman measurements. HB provided the Polly[XT] results. DA provided technical support for the BERTHA lidar system. CT and BT provided the six and seven wavelength GRASP retrieval. MR did the source appointment analysis. UW provided comments on light scattering.

*Competing interests.* The authors declare that they have no conflict of interest.

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

**Table 1.** Intensive optical properties of Saharan dust observed at Leipzig on 22–23 February (pure dust case) and 3 March 2021 (slightly polluted dust case). For comparison, the optical properties of long-range-transported Saharan dust at Barbados as observed during the SALTRACE campaign are provided, the PLDR during SALTRACE are taken from Haarig et al. (2017). The PLDR of the Leipzig cases are reported for the 3-wavelength depolarization setup of the measurement (first 20 min), whereas the other properties are reported for the 3-wavelength extinction measurement. SAL – Saharan Air layer, PLDR – particle linear depolarization ratio, EAE – extinction Ångström exponent, BAE – backscatter Ångström exponent

|  | Wvl. (nm) | 22–23 Feb 2021 | 3 March 2021 | SALTRACE |
|---|---|---|---|---|
| Height range |  | 3–5 km | 2–3.5 km | SAL |
| Lidar ratio | 355 | 47±8 sr | 49±4 sr | 59±16 sr |
|  | 532 | 50±5 sr | 46±5 sr | 57±8 sr |
|  | 1064 | 69±14 sr | 57±9 sr | 67±15 sr[a] |
| Ratio of lidar ratios | 355/532 | 0.94±0.19 | 1.07±0.14 | 1.04±0.30 |
|  | 1064/532 | 1.38±0.31 | 1.24±0.24 | 1.31±0.32[b] |
| PLDR | 355 | 0.242±0.024 | 0.174±0.041 | 0.252±0.030 |
|  | 532 | 0.299±0.018 | 0.292±0.016 | 0.280±0.020 |
|  | 1064 | 0.206±0.010 | 0.242±0.007 | 0.225±0.022 |
| Ratio of PLDR | 355/532 | 0.81±0.09 | 0.60±0.14 | 0.90±0.12 |
|  | 1064/532 | 0.69±0.05 | 0.83±0.05 | 0.80±0.10 |
| EAE | 355/532 | –0.005±0.186 [c] | 0.332±0.051 | 0.103±0.254 |
|  | 532/1064 | –0.083±0.213 | 0.008±0.079 | – |
|  | 355/1064 | –0.098±0.139 | 0.128±0.067 | – |
| BAE | 355/532 | 0.040±0.524[c] | 0.120±0.276 | -0.030±0.153 |
|  | 532/1064 | 0.347±0.260 | 0.345±0.260 | 0.474±0.201 |
|  | 355/1064 | 0.187±0.193 | 0.262±0.144 | – |

[a] Estimated using lidar and AERONET.

[b] Here the lidar ratio at 532 nm (51 sr) of the same day (20 June 2014) was used.

[c] Polly$^{XT}$ data at 355 and 532 nm were used.

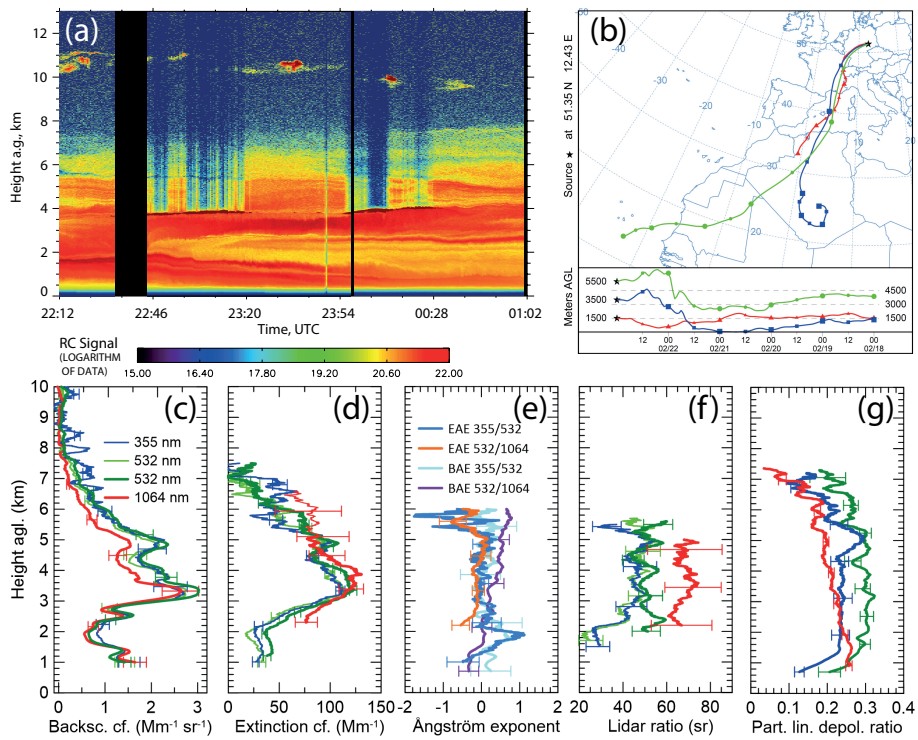

**Figure 1.** Saharan dust observations at Leipzig on 22–23 February 2021, 22:12–01:02 UTC, (a) time-height display of the 532 nm cross-polarized, range-corrected signal. (b) HYSPLIT backtrajectory for 23 February 2021, 00:00 UTC (<2 days from the Sahara to Leipzig). (c) Particle backscatter coefficient (200 m vertical smoothing), (d) particle extinction coefficient (950 m vertical smoothing, 2000 m (3290 m) at 1064 nm below (above) 5 km), (e) extinction Ångström exponent (EAE) and backscatter Ångström exponent (BAE) for the given wavelengths pairs, 355/532 from Polly[XT], 532/1064 from BERTHA, (f) lidar ratio (950 m vertical smoothing, 2000 m at 1064 nm), (f) particle linear depolarization ratio (200 m vertical smoothing, 750 m at 355 nm). BERTHA provided measurements at 532 nm (dark green) and 1064 nm (red) and Polly[XT] at 355 nm (blue) and 532 nm (light green). All optical properties are shown for the period 22:45–01:02 UTC, except the depolarization ratios are shown 22:12–22:32 UTC to be at the same time as the 1064 nm depolarization ratio measurement.

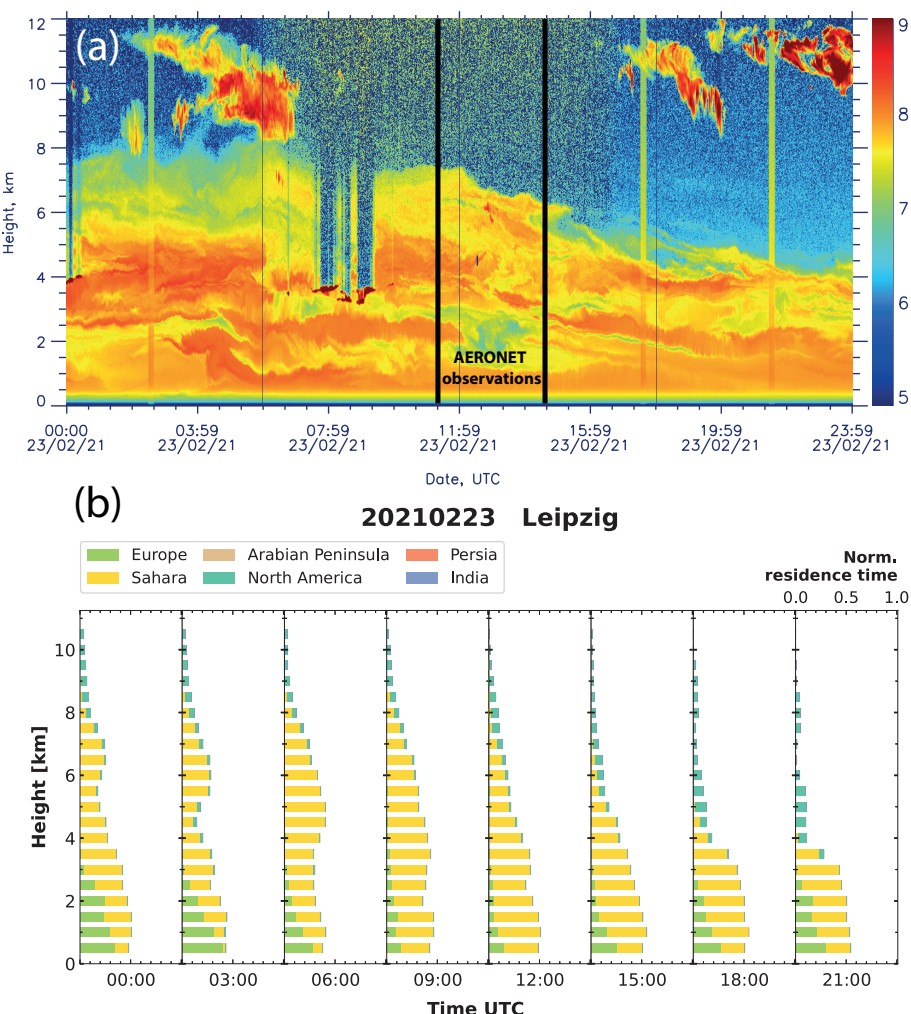

**Figure 2.** Development of the dust layer over Leipzig on 23 February 2021. (a) Cross-polarized, range-corrected signal at 532 nm measured with Polly[XT]. The period of the five AERONET observations is marked by thick black vertical bars. (b) Source attribution in 3-hour intervals by using the method of Radenz et al. (2021). The normalized residence time of the air masses close to the ground (below 2 km height) within 10 days prior to the arrival over Leipzig at the indicated time stamps on 23 February 2021 is shown. The trajectory calculations are based on FLEXPART. The colors indicate different regions defined in Radenz et al. (2021). Here, the Saharan desert, continental Europe and North America are relevant.

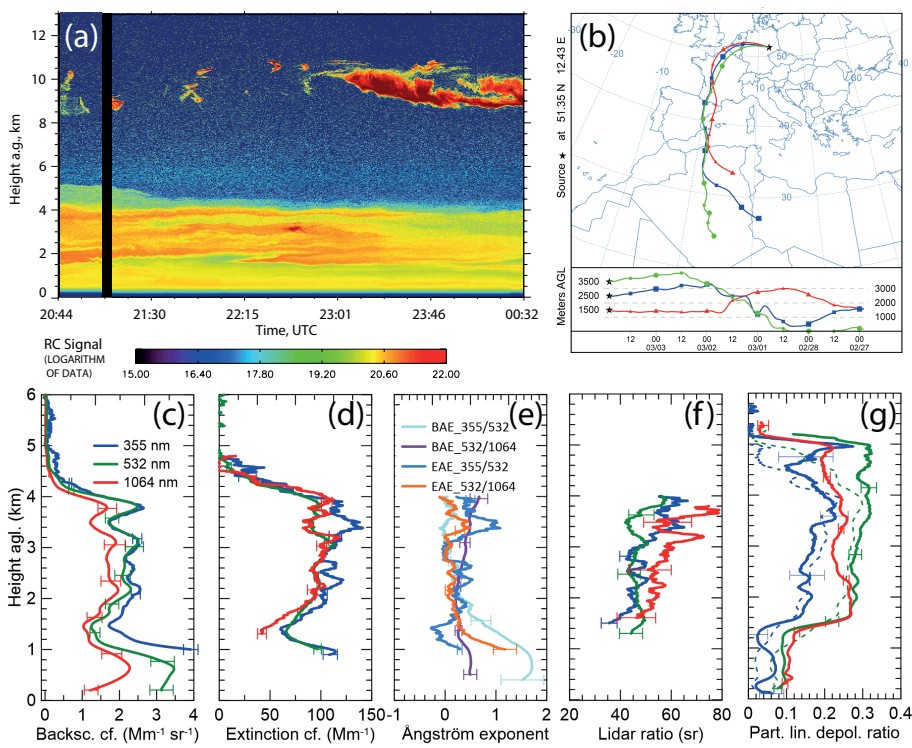

**Figure 3.** Same as Fig. 1, except for 3-4 March 2021, 20:44–00:32 UTC, (a) time-height display of the 532 nm cross-polarized, range-corrected signal. (b) HYSPLIT backtrajectory for 3 March 2021, 22:00 UTC. (c) Particle backscatter coefficient (200 m vertical smoothing), (d) particle extinction coefficient (750 m vertical smoothing), (e) extinction Ångström exponent (EAE, 750 m vertical smoothing) and backscatter Ångström exponent (BAE, 200 m vertical smoothing) for the given wavelengths pairs, (f) lidar ratio (750 m vertical smoothing), (g) particle linear depolarization ratio PLDR (200 m vertical smoothing). All shown optical properties are the mean profiles from 21:11–00:32 UTC, except the thick lines of PLDR (20:44–21:04 UTC). The dashed lines of the PLDR at 355 and 532 nm belong to the main measurement at 21:11–00:32 UTC, where the dust layer height decreased.

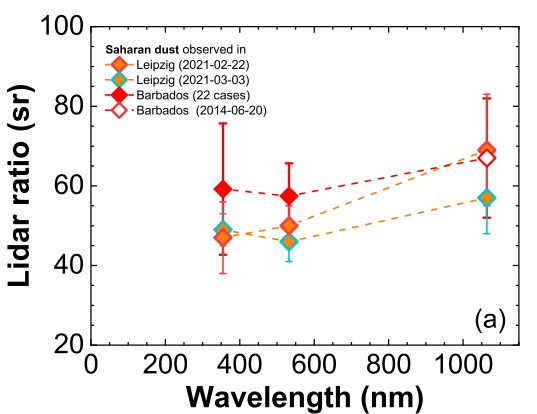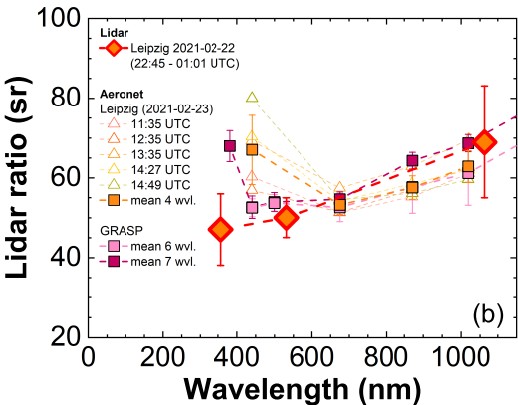

**Figure 4.** (a) Spectral dependence of the lidar ratio of Saharan dust observed at Leipzig and Barbados (during SALTRACE). The lidar ratio at 1064 nm during SALTRACE (open diamond) was estimated on 20 June 2014 combining AERONET AOD and lidar observations (see text for explanations and Mamouri and Ansmann (2017)). (b) Spectral dependence of the lidar ratio of Saharan dust observed in the night of 22–23 February 2021 compared with AERONET v3 inversion solutions for the lidar ratio on 23 February 2021. The results of the five photometer scans (open triangles) at indicated times are shown together with the mean of the four-wavelength retrieval (orange squares, standard retrieval) and the six and seven wavelength retrieval (pink and purple squares, GRASP method). The GRASP retrieval at 1640 nm (not shown) leads to lidar ratios of 94±8 (using six wavelengths) and 101±4 sr (using seven wavelengths) for this long wavelength.

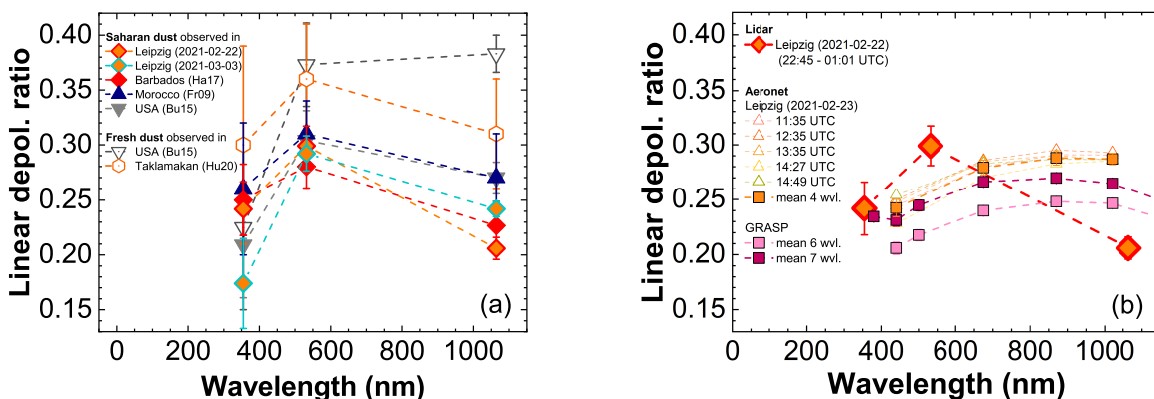

**Figure 5.** (a) Spectral dependence of the particle linear depolarization ratio (PLDR) of desert dust. The Leipzig observations are compared with triple-wavelength depolarization-ratio measurements of Saharan dust and fresh dust available in the literature (Ha17 – Haarig et al. (2017), Fr09 – Freudenthaler et al. (2009), Bu15 – Burton et al. (2015), Hu20 – Hu et al. (2020)). (b) Spectral dependence of the PLDR of Saharan dust observed in the night of 22–23 February 2021 compared with AERONET v3 inversion solutions for the PLDR on 23 February 2021. The results of the five photometer scans at indicated times are shown together with the mean of the four wavelength retrieval (orange squares, standard retrieval) and the six and seven wavelength retrieval (pink and purple squares, GRASP method). The GRASP retrieval at 1640 nm (not shown) leads to PLDR values of 0.184±0.007 (using six wavelengths) and 0.192±0.008 (using seven wavelengths) for this long wavelength.