# Peer review of "First triple-wavelength lidar observations of depolarization and extinction-to-backscatter ratios of Saharan dust"

_Atmospheric Chemistry and Physics, 2021_

## Referee Comment (RC2)

Review of "First triple-wavelength lidar observations of depolarization and extinction-tobackscatter ratios of Saharan dust" by M. Haarig, A. Ansmann, R. Engelmann, H. Baars, D. Althausen, C. Toledano, B. Torres, M. Radenz, and U. Wandinger

reviewed by Mark Vaughan <mark.a.vaughan@nasa.gov>

In this manuscript, the authors use approximately 6 hours of measurements acquired on 22–23 February and 3 March 2021 to characterize the lidar ratios and particulate depolarization ratios retrieved from three wavelength Raman lidar measurements of Saharan dust transported over Leipzig, Germany. They then compare the lidar-derived parameters to the same quantities estimated using the GRASP algorithm and AERONET measurements acquired ~12 hours prior to the February lidar measurements. From these comparisons they conclude that the AERONET/GRASP technique (a) adequately reproduces the spectral dependence of Saharan dust lidar ratios at longer wavelengths (675 nm and above), but fails at shorter wavelengths (440 nm); and (b) deficiencies remain in reproducing the spectral dependence of the particulate depolarization ratio.

This is a well-organized and well written manuscript that I expect will be of particular interest for space-based aerosol lidar applications. Subject to the caveat initially put forward by the content editor (i.e., this manuscript should be published as a "measurement report", not as a research article), the topic is well-suited for publication in ACP. To my mind, the primary contribution of this work lies in providing retrievals of Saharan dust lidar ratios at 1064 nm. The lidar literature abounds with reports of retrievals of dust lidar ratios from across the globe at 355 nm and, to a somewhat lesser extent, at 532 nm. But to my knowledge, BERTHA is unique in being able to deliver high quality *range-resolved* retrievals of 1064 nm lidar ratios. Nevertheless, I have a few quibbles about some of the content that prevent me from fully endorsing the manuscript in its current form. I outline these in a few brief paragraphs below and in an annotated version of the manuscript attached at the end of this review. Once the authors have considered these remarks, I will gladly recommend publication of their paper.

**General Remarks**

While the manuscript briefly touches on the technique used for the polarimetric calibration (lines 78–80), I find no mention of the method(s) used for the radiometric calibration of the 1064 nm channel (or of any of the other channels, for that matter). The paper should be expanded to include a sentence or two describing the most prominent contributors to the retrieval uncertainties at all three wavelengths. While I understand that "more details can be found in Haarig et al. (2016)" (line 71), the cirrus scenes examined in that paper offer relatively straightforward calibration opportunities, and the dust cases in this paper could be more challenging.

I'm not sure I understand what the authors are showing in panel (f) of Figures 1. They quite clearly state that the 1064 nm depolarization ratio measurements were conducted only during the first 20 minutes of the observation period. So in Figure 1(f), are the 355 nm and 532 nm averaged profiles also restricted to data acquired over the first 20 minutes? If so, the text and caption should both clearly state this fact. And if not, the figure should be recreated using only the first 20 minutes of the 355 nm and 532 nm measurements. Similarly, are the data in the averaged profiles shown in panels (c), (d), and (e) restricted to observations acquired between 22:45 and 01:02 UTC? Again, if they are, please say so unambiguously; and if they are not, please recreate the plots so that all data are from the exact same time period. These comments also apply to panels (c), (d), (e), and (f) in Figure 3.

**First triple-wavelength lidar observations of depolarization and extinction-to-backscatter ratios of Saharan dust**

Moritz Haarig1, Albert Ansmann1, Ronny Engelmann1, Holger Baars1, Dietrich Althausen1, Carlos Toledano2, Benjamin Torres3, Martin Radenz1, and Ulla Wandinger1

1Leibniz Institute for Tropospheric Research, Leipzig, Germany

2Atmospheric Optics Group, University of Valladolid, Valladolid, Spain

[revised manuscript text omitted]

---

## Author Comment (AC1)

**Dear anonymous referee,**

**Thank you very much for your time and effort you spend on reviewing our manuscript. We are grateful to your detailed comments and suggestions. And we did our best to incorporate them into the manuscript.**

**First, we want to introduce the major changes on the manuscript:**

1. **Motivated by your comments, we checked the cross talk from the elastic wavelength in the 1058 nm Raman channel. Indeed, a cross talk was present and has been corrected (see below). Furthermore, a temperature correction was applied.**
2. **In the discussion, a dedicated section for the comparison of the AOD and Ångström exponents was introduced. Furthermore, the vertical profiles of the extinction and backscatter Ångström exponent were included in Figure 1 + 3.**
3. **We strengthened the point, that the columnar photometer results include the fine-mode pollution in the planetary boundary layer, whereas the lidar results focus on the Saharan dust layer.**

**Now, we respond in bold to your specific comments. Attached, you'll find the revised version of the manuscript with changes indicated in bold.**

Concerning the technique for measuring the 1064nm extinction coefficient

*Two interference filters (IFF) with a bandwidth of 9nm are placed one after the other before the infrared Raman channel in order to suppress the light from the elastic scattering. Even though the transmission information at 1064nm from the manufacturer seems sufficient for this, factors such as the temperature and/or the incidence angle of the collected light on the IFF could shift the transmission spectrum of the IFF allowing cross-talks from the elastic line. Have the authors checked that this is not the case experimentally? This can be checked by placing a third typical IFF centered at 1064nm in the rotational Raman channel and see if any significant amount of light is detected.*

**We are thankful for the reviewer's suggestions. Unfortunately, we are not able to check the cross talk experimentally as suggested, because the two Raman filters are currently installed in our new system at Cabo Verde. New filters are ordered but haven't arrived yet. Nevertheless, we investigated the issue of possible cross-talk using the water clouds at around 4 km height in the measurement of 22 February 2021 (case study 1). And indeed, we found some signal from the clouds in our 1058 Raman channel. Thus, we subtracted the elastic signal (1064 nm) times a cross -talk factor from our Raman signal (1058). We were iterating the cross-talk factor until the influence of the clouds vanished from our Raman signal at cloud base. We could not use the center of the cloud because the elastic signal was saturated there. Additionally, we scaled the 607 nm Raman signal to check the slope of the decreasing 1058 nm Raman signal in the first 100 m of the cloud. We found a cross-talk factor of 0.0175±0.0005, so 1.75% of our elastic signal were detected in the Raman channel. We implemented the cross-talk correction of the 1058 nm Raman signal in our data processing and added the following statement to the manuscript (lines 84-88):**

**"However, still some cross-talk from the elastic signal at 1064 nm was present in the Raman channel. The enhanced signal received from the liquid clouds on 22 February 2021 (Section 3.1) was used to estimate the cross-talk factor of 0.0175±0.0005. For the further analysis, the elastic signal at 1064 nm**

**multiplied by the cross-talk factor was subtracted from the Raman signal at 1058 nm to correct for the cross talk. In this simple approach, the cross talk is assumed to be temperature independent."**

**We are grateful for the reviewer's comment, because the cross-talk correction is important for a proper extinction measurement.**

*In addition, the Raman spectrum is affected by the air temperature. Is the bandwidth of the IFF sufficient to eliminate any temperature effects in the profiles? What are the uncertainties? Even if such issues are reported in a previous study, a reference must be added along with a few lines explaining the main findings.*

**Thank you for pointing to the temperature effect. The previous study (Haarig et al., AMT 2016) stated: "The detected Raman backscatter intensity is only weakly temperature dependent (4% increase of the measured rotational Raman signal for a temperature decrease from 300 to 230 K)." However, we calculated the temperature dependence of our Raman filters, again, using the temperature range relevant for the present study. We found that the temperature dependent transmission of the oxygen and nitrogen Raman lines varies from 0.729 to 0.793 in the temperature range (288-208 K) observed on 22 February 2021 in the troposphere. We corrected the 1058 nm Raman signal for the temperature dependence and found around 3 % higher extinction values in the dust layer for the temperature corrected signal. For more detailed information about the temperature dependence of the rotational Raman signals we refer to Veselovskii et al., 2015. In the manuscript we added the following statement (lines 88-93):**

**"The temperature dependent transmission of the oxygen and nitrogen Raman lines varies from 0.729 to 0.793 between 288 and 208 K, which was the maximum and minimum temperature observed in the troposphere on 22 February 2021. The rotational Raman signal at 1058 nm was corrected for the temperature dependence after the cross-talk correction. Within the dust layer, the correction led to around 3% higher extinction values. Detailed calculations of the temperature dependence of rotational Raman lidar signals can be found in Veselovskii et al. (2015)."**

**Thank you for your helpful comment. We started the effort to calculate the temperature dependence of all our interference filters in the other systems, especially for the vibrational-rotational Raman channels and the depolarization channels.**

*Temperature effects can be even more pronounced in the near range (below the full overlap region) where the angle of incidence of the collected light in the IFF is still expected to change with range, especially for biaxial systems. This translates to a shift of the central wavelength of the IFF which in turn could result to temperature dependencies, especially if the IFF central wavelength shifts to a more temperature sensitive region. In the near range this could create overlap-like effects to the signal. Values of the extinction profiles at 1064nm in figures 3 and 4 are not provided in the near range, probably due to overlap issues. Could temperature issues be the reason why the 1064nm extinction profile in figure 3 starts at 2km while the 355nm and 532nm profiles start at 1km? Please add a description explaining whether such issues are present. If they are, please provide some information on how they have been dealt with.*

**This is a valid point and needs further investigation. For further studies, we will investigate the influence of the angle of incidence on the near range signal. For the current publication, we decided to not use the near range.**

**Additionally, the long vertical smoothing length of 2000 m (267 bins) in the case of the 1064 nm extinction on 22 February led to the decision to not show the profiles below 2 km height. At 355 and 532 nm a vertical smoothing of 950 m (127 bins) was sufficient, enabling us to show the profile down to 1 km height. On 3 March 750 m vertical smoothing for all profiles were sufficient because of the longer temporal averaging. It is not stated in the manuscript, but BERTHA is a coaxial system in contrast to our Polly lidars.**

Concerning the comparisons with the AERONET inversions

*The lidar and sunphotometer measurements are not simultaneous. They have a time difference that is larger than 11 hours (around 0 UCT for the lidar -middle time -, and 11:35 UTC for the first sunphotometer measurement). The atmosphere could have changed significantly in the meantime. The authors use only the inversion on 23rd of February in their analysis. The retrievals on 22 of February are also available. They should be used to indicate the degree of atmospheric change in Figure 4, Figure 5, and in the discussion. Looking at the AERONET size distributions on 22 and 23 it seems that there are substantial changes. Furthermore, the inversion on 3rd of March is also available. Having provided the degree of atmospheric change in a ~11hour interval from the 22/02 case, the authors could add the 2 inversions on 03/03 in the analysis since the time difference is not far greater (~14 hours).*

**Our aim was to compare the Saharan dust events from lidar and sun photometer observations. We have chosen the closest sun photometer observations of Saharan dust to our lidar measurements. Indeed, there are further photometer observations available for 22 February (until 15:30 UTC). The first slices of Saharan dust arrived at around 16 UTC over Leipzig, the full dust layer at around 19 UTC. The development of the Saharan dust plume can be studied using the continuous observations from PollyNET ([https://polly.tropos.de/calendar/location/1](https://polly.tropos.de/calendar/location/1)). We decided to not use the photometer observations from 22 February as they don't measure the Saharan dust plume. However, we calculated the mean of the photometer observations of 23 – 25 February, where the Saharan dust layer was continuously present. The values were similar, but the agreement was a slightly better for the 23 February 2021. Thus, we decided to not use them for the publication. Concerning the results of 3 March 2021, there are no photometer observations of the Saharan dust layer. On 3 March, together with the Saharan dust plume cirrus clouds were present over Leipzig preventing photometer observations. The rain event on 4 March around 6 UTC, removed the dust. Again, no photometer observations of the Saharan dust plume were possible. Again, we used the continuous observations from PollyNET to study the development of the dust layer.**
**Now, it is stated in the paper in a clearer way, why we don't use further photometer measurements for comparison. See lines 163-165 and 190-192:**

**"The AERONET measurements from 22 February 2021 did not capture the Saharan dust event, which arrived at 19 UTC at Leipzig, first slices already at 16 UTC."**

**"Unfortunately, no AERONET observations could be used for comparison. The dust layer was accompanied by extended cirrus layers preventing sun photometer observations and on the next day the precipitation event removed the dust."**

*In addition, an AOD and Angstrom exponent comparison part or section should be*

*included in the manuscript prior to the inversion comparisons in the Discussion as these AERONET products are far more accurate. Extinction and backscatter-related Angstrom profiles are also missing from figures 3 and 4. Please include them as well.*

**Following your suggestion, we introduced a new chapter 4.1 "Backscatter and extinction Ångström exponent" to compare the AOD and EAE to the AERONET observations for 22 February 2021 and the BAE to literature values. Furthermore, we added in Fig. 1 + 3 vertical profiles of the BAE and EAE for the wavelength pairs of 355/532 and 532/1064. These additions make indeed the comparison more complete.**

*Moreover, it should be stressed in the Discussion section that differences with the AERONET retrievals, especially in the lidar ratio, are expected also due to the overlap function of the lidar system. For figure 1 and 3 it is obvious that the extinction profiles are available only above 1km and the 1064nm extinction profile above 2km! This is certainly not negligible, and could affect the comparison, even for the intensive properties since the aerosol type tends to be different inside the PBL. As mentioned above, a comparison with the AERONET AOD and Angstrom values will help quantify the degree of the uncertainties in the overlap region.*

**We improved the manuscript in this point. Firstly, we calculated the AOD from the lidar measurements. For the overlap region, we had to assume the backscatter coefficient times a lidar ratio of 50 sr. To assess the uncertainty, we varied the lidar ratio from 40 to 60 sr. Secondly, we strengthened the discussion about the fine-mode pollution in the boundary layer which certainly affect the columnar photometer observations, especially at the shorter wavelengths. The enhanced lidar ratios in the UV are most probably caused by the influence of the PBL. Whereas the lidar-derived values correspond to the dust layer only.**

Concerning the 1064 depolarization ratio profiles

*The 1064nm PLDR profiles have different vertical structure than the 355nm and 532nm profiles which in general show similar patterns. This is especially visible in figure 1 where PLDR maxima and minima are reproduced in both 355nm and 532nm profiles but not in the 1064nm profile. Can the authors explain this behavior? Is this a matter of atmospheric variability since the 1064nm depol. measurements last only for the first 20 minutes. In this case, and serving mainly as a test, would the profiles be more consistent if the first 20 minutes are used so that all thee depolarization ratios can be measured at the same time?*

**Yes, you are right. We changed Figure 1 + 3 to show all depolarization ratio profiles at the same time (first 20 min). In Figure 3, we show additionally the PLDR profiles at 355 and 532 nm for the longer measurement as dashed lines. Furthermore, in Table 1 we are just using the depolarization ratios measured simultaneously during the first 20 min.**

Technical Comments
*-- Line 111-112: Have the two instruments been intercompared? Please provided references.*

**We have done previous comparisons of BERTHA and Arielle PollyXT measurements, but there are no publications about it. BERTHA was compared to POLIS measurements at Barbados (Haarig et al., 2017) and to another PollyXT for the stratospheric Canadian smoke event (Haarig et al., 2018). We add some more technical details about Arielle PollyXT.**

*-- Line 114, Lines 144-145: Here it is not clear what the reference value is and how the cirrus clouds facilitate the selection of a better value, especially for people outside of the lidar community. The authors should provide a clearer description and provide references.*

**We moved the description of the backscatter calibration to the Instruments section according to a comment by another reviewer (lines 69-75). Therefore, we removed the sentences from the description of the case study. Indeed, we found that the backscatter coefficient in the cirrus cloud was not wavelength independent as we previously assumed. The thick cirrus on 3 March had a backscatter color ratio (1064/532) of 0.88. The cirrus cloud on 22 February was too thin to draw firm conclusions.**

*-- Figure 1, 3, and the tables: How were the mean lidar ratio and Angstrom values calculated? From averaging the lidar ratio profile or by averaging the extensive products (backscatter, extinction) first and then dividing them? If not yet applied, the authors should switch to the later as this leads to less noise.*

**We have done it in both ways and the lidar ratio has not changed. The same vertical averaging was applied for extinction and backscatter coefficients to derive the lidar ratio. In case of the Ångström exponents, the results improved when taking the average of extensive products.**

*-- Figure 4 and 5 on the right: Please add also the time of the lidar measurement in the Figures.*

**Thank you, the time has been added.**

[revised manuscript text omitted]

---

## Author Comment (AC2)

**Dear Mark Vaughan,**

**Thank you very much for your time and effort you spend on reviewing our manuscript. We are grateful to your detailed comments and suggestions. And we did our best to incorporate them into the manuscript.**

**First, we want to introduce the major changes on the manuscript:**

1. **Motivated by the comments of referee #1, we checked the cross talk from the elastic wavelength in the 1058 nm Raman channel. Indeed, a cross talk was present and has been corrected (see below). Furthermore, a temperature correction was applied.**
2. **In the discussion, a dedicated section for the comparison of the AOD and Ångström exponents was introduced. Furthermore, the vertical profiles of the extinction and backscatter Ångström exponent were included in Figure 1 + 3.**
3. **We strengthened the point, that the columnar photometer results include the fine-mode pollution in the planetary boundary layer, whereas the lidar results focus on the Saharan dust layer.**

**For the discussion. I copied your comments (in italic) from the pdf. Our responses are given in bold. Attached, you'll find the revised version of the manuscript with changes indicated in bold.**

*In this manuscript, the authors use approximately 6 hours of measurements acquired on 22–23 February and 3 March 2021 to characterize the lidar ratios and particulate depolarization ratios retrieved from three wavelength Raman lidar measurements of Saharan dust transported over Leipzig, Germany. They then compare the lidar-derived parameters to the same quantities estimated using the GRASP algorithm and AERONET measurements acquired ~12 hours prior to the February lidar measurements. From these comparisons they conclude that the AERONET/GRASP technique (a) adequately reproduces the spectral dependence of Saharan dust lidar ratios at longer wavelengths (675 nm and above), but fails at shorter wavelengths (440 nm); and (b) deficiencies remain in reproducing the spectral dependence of the particulate depolarization ratio. This is a well-organized and well written manuscript that I expect will be of particular interest for space-based aerosol lidar applications. Subject to the caveat initially put forward by the content editor (i.e., this manuscript should be published as a "measurement report", not as a research article), the topic is well-suited for publication in ACP. To my mind, the primary contribution of this work lies in providing retrievals of Saharan dust lidar ratios at 1064 nm. The lidar literature abounds with reports of retrievals of dust lidar ratios from across the globe at 355 nm and, to a somewhat lesser extent, at 532 nm. But to my knowledge, BERTHA is unique in being able to deliver high quality range-resolved retrievals of 1064 nm lidar ratios. Nevertheless, I have a few quibbles about some of the content that prevent me from fully endorsing the manuscript in its current form. I outline these in a few brief paragraphs below and in an annotated version of the manuscript attached at the end of this review. Once the authors have considered these remarks, I will gladly recommend publication of their paper.*

**Thank you for your kind words and the support of our opinion that the current manuscript is a research article and not a measurement report.**

*General Remarks*
*While the manuscript briefly touches on the technique used for the polarimetric calibration (lines 78–80), I find no mention of the method(s) used for the radiometric calibration of the 1064 nm channel (or of any of the other channels, for that matter). The paper should be expanded to include a sentence or two describing the most prominent*

*contributors to the retrieval uncertainties at all three wavelengths. While I understand that "more details can be found in Haarig et al. (2016)" (line 71), the cirrus scenes examined in that paper offer relatively straightforward calibration opportunities, and the dust cases in this paper could be more challenging.*

**We added some sentences to describe the radiometric calibration (Rayleigh calibration) in the instrument section. Indeed, the backscatter coefficient was not wavelength independent in the cirrus cloud as originally expected.**
**"The radiometric or Rayleigh calibration of the signals was done in clear air which is at 1064 nm a challenging task. At 1064 nm the elastic return signal from molecules is 81 times and 16 times smaller than for 355 and 532 nm, respectively. The reference height for the 1064 nm signal was set some hundreds of meters above the dust layer top height for a sufficient strong molecular signal. On 3 March 2021 (Sect. 3.2) a thick cirrus cloud was present, which could be used to check the Rayleigh calibration. A backscatter color ratio (1064/532) of 0.88 was observed, which is in line with observations by Vaughan et al. (2010) who reported values of 1.01±0.25 in cirrus clouds. On 22 February 2021 (Sect. 3.1), the cirrus clouds were too thin to check the calibration."**

*I'm not sure I understand what the authors are showing in panel (f) of Figures 1. They quite clearly state that the 1064 nm depolarization ratio measurements were conducted only during the first 20 minutes of the observation period. So in Figure 1(f), are the 355 nm and 532 nm averaged profiles also restricted to data acquired over the first 20 minutes? If so, the text and caption should both clearly state this fact. And if not, the figure should be recreated using only the first 20 minutes of the 355 nm and 532 nm measurements. Similarly, are the data in the averaged profiles shown in panels (c), (d), and (e) restricted to observations acquired between 22:45 and 01:02 UTC? Again, if they are, please say so unambiguously; and if they are not, please recreate the plots so that all data are from the exact same time period. These comments also apply to panels (c), (d), (e), and (f) in Figure 3.*

**Thank you for your comment. We recreated Figure 1 + 3 in the way to show all optical profiles from the long measurement (3+3+2 configuration), only the profiles of the PLDR are shown from the short measurement during the first 20 min (3+2+3 configuration). In that way all PLDR profiles are from coincident measurements. In Table 1 we now provide the PLDR from the first 20 minutes to have them measured all at the same time.**
**On 22 February, the vertical structure of the dust layer did not change between the measurements, except of the lowest 2 km, there the depolarization ratio decreased. This behavior could be seen in the time-height plot of the cross-polarized signal (Fig. 1a). In Fig. 3 we added the PLDR profiles at 355 and 532 nm from the long measurement as dashed lines. The dust layer descended, but the mean values in the 2.0-3.5 km height range did not changed.**
**We could not show all profiles from same time, as we can either measure the depolarization ratio or the extinction coefficient at 1064nm.**
**Our statement in the Figure caption now clearly states for which time period the profiles are shown.**

*New information at 1064 should be especially helpful in interpreting measurements made by the Cloud Aerosol Transport System (CATS) lidar (http://cats.gsfc.nasa.gov/), as the vast majority of the CATS measurements were at 1064 nm only.*
*Also note that NASA is planning to launch to new space-based Earth-observing lidars in the coming years (see https://vac.gsfc.nasa.gov/accp/home.htm), and that both of these will be outfitted with an elastic backscatter channel at 1064 nm. beefing up our collective knowledge about 1064 nm lidar ratios should be a huge help in deriving the best possible estimates of aerosol optical properties from these future systems.*

**Thank you for mentioning the importance for CATS and future NASA missions. We included it in the introduction.**

*Please state the wavelengths measured by Arielle PollyXT and the measurement technique used (e.g., Raman, HSRL, elastic backscatter). in particular, I'd like to get some understanding about how the Arielle PollyXT data augments/complements the BERTHA data \*without\* having to review Engelmann et al., 2016.*

**We included a short description with the key facts of PollyXT in the instrument section.**
**"It is a Raman polarization lidar, measuring the backscatter coefficient at 355, 532 and 1064 nm, the extinction coefficient at 355 and 532 nm (vibrational-rotational Raman) and the depolarization ratio at 355 and 532 nm. Additionally, near range channels at 355, 387, 532 and 607 nm provide information at lower altitudes and the dual-field-of-view polarization technique (Jimenez et al., 2020) is implemented to study clouds."**

*Why? an instrument malfunction maybe? some additional info/context would be helpful.*

**We have two lasers, one emitting at 532 nm, the other at 355 and 1064 nm. The second laser had some problems with emitting reduced power at 355 nm. The neutral density filters were not adapted sufficiently to the reduced power. This issue was solved until the next measurement. A statement was added in Section 3.1:**
**"The signals at 355 nm of the BERTHA lidar were very weak during that night. In fact, neutral density filters in front of the UV channels were not adapted to the strongly reduced power of the UV laser."**

*Voudouri et al., 2020 recently published evidence to the contrary. so perhaps this assertion is true only for a subset of cirrus? what are the backscatter color ratios (or backscatter-related Angstrom exponents) for the cirrus observed during these dust measurements? these values should be included in the manuscript.*

**You are right. We revised the Rayleigh calibration. The cirrus cloud on 22 February was too thin to check the calibration. But, the thick cirrus on 3 March was strong enough. A backscatter color (1064/532) ratio of 0.88 was observed. We reported this value in the instruments section. However, we decided to not report the backscatter and extinction Ångström exponents for the cirrus cloud because this publication focuses on the dust observations and not on cirrus observations.**

*This is very curious wording. in the first outbreak, data were averaged over 2 hours and 15 minutes. does this shorter averaging interval mean that the extinction values retrieved at 1064 nm are not trustworthy?*

**The word "trustworthy" was misleading. We rephrased the sentence to:**
**"To avoid too strong vertical smoothing for the extinction coefficient at 1064 nm compared to the previous case, signal profiles collected over 3 h 20 min (21:11–00:32 UTC) were averaged.**

*Please explain why different sliding window lengths are used for the two sets of measurements*

**2h15min of data were collected for the first measurement, 3h20min for the second measurement. From the 2h15min, we had to exclude the cloudy profiles. In the end, the profiles of the 1064 nm extinction were very noisy on 22 February, so we had to increase the vertical smoothing. Also, for the other**

**wavelengths, we could reduce the vertical smoothing for the second dust event with 3h20min of temporal averaging. We included a statement in Sect. 3.2: "To avoid too strong vertical smoothing for the extinction coefficient at 1064 nm compared to the previous case, signal profiles collected over 3 h 20 min (21:11–00:32 UTC) were averaged."**

*Very interesting. the magnitude of the difference in lidar ratios is similar to what was reported by Liu et al. (2008) using CALIPSO measurements. given the very substantial changes to the CALIOP 1064 nm calibration coefficients made since 2008, it's not entirely clear that a comparison between the two sets of results would be really relevant. but it remains interesting nevertheless.*

**Thank you for pointing us to the study of Liu et al., 2008. We included a sentence about the increase of the lidar ratio from 532 to 1064 nm found by Liu et al.**

*Why is it obvious that the difference in PLDR should be attributed solely to mixing with anthropogenic pollution and not to potential differences in the mineralogy of the dust source regions? lidar ratios vary considerably with source region (e.g., Schuster et al., 2012). it does not seem inconceivable that depolarization ratios could vary as well.*

**Over the past years and a lot of field campaigns in Morocco, Cabo Verde, Barbados, Cyprus, Israel, Tajikistan, Greece, we concluded that a depolarization ratio of around 0.30, at least at 532 nm is a characteristic feature of mineral dust and does not vary with source region. The depolarization ratio at 355 nm is more sensitive to mixing with fine-mode pollution. And for the depolarization ratio at 1064 we definitely need more observations, but we have indications that it is quite sensitive to transport way and the removal of larger dust particles.**
**The dust – non-dust separation schemes by Shimizu et al., 2004, Tesche et al., 2009 and Mamouri and Ansmann, 2017 are based on the assumption of a somehow universal value of the dust depolarization ratio of around 0.3 at 532 nm.**
**In contrast, the lidar ratio depends on source region as shown by Schuster et al., 2012 and Hofer et al., 2020.**

*This "clear evidence" is quite sparse (two events) and applicable to Saharan dust only.*

**We changed the sentence to: "The present study indicates an increase of the lidar ratio from 532 to 1064 nm."**

*The SAMUM study measured pristine, freshly mobilized dust. on the other hand, the current study focuses on aged dust that has been transported over a considerable distance and has mostly likely been mixed with other aerosols of anthropogenic origin. it should be no surprise then that different measurement conditions yield different results.*

**We agree and added a statement on line 253: "In contrast to those measurements for freshly emitted dust close to its source, we found now lidar ratio values of 47±8 sr, 50±5 sr, and 255 69±14 sr for the three wavelengths after a dust transport of less than 2 days."**

*I don't understand the use of the word "drop"? according to Shin et al., the AERONET retrievals show "an increase in the imaginary part of the refractive index with decreasing wavelength". So, I believe this sentence should be reworded to say "The increase in the imaginary part of the refractive index at 440 nm compared to 675 nm is too large in the AERONET inversion procedure..."*

**You are right. We corrected the sentence accordingly. Thank you.**

*The Schuster study found significant differences in the real part of the refractive index, not the imaginary part.*

**Thank you for the comment. The manuscript was changed accordingly.**

[revised manuscript text omitted]

---

## Author Comment (AC3)

**Dear Oleg Dubovik,**

**Thank you very much for your time and effort you spend on reviewing our manuscript. We are grateful to your detailed comments and suggestions. And we did our best to incorporate them into the manuscript.**

**First, we want to introduce the major changes on the manuscript:**

1. **Motivated by the comments of referee #1, we checked the cross talk from the elastic wavelength in the 1058 nm Raman channel. Indeed, a cross talk was present and has been corrected (see below). Furthermore, a temperature correction was applied.**
2. **In the discussion, a dedicated section for the comparison of the AOD and Ångström exponents was introduced. Furthermore, the vertical profiles of the extinction and backscatter Ångström exponent were included in Figure 1 + 3.**
3. **We strengthened the point, that the columnar photometer results include the fine-mode pollution in the planetary boundary layer, whereas the lidar results focus on the Saharan dust layer.**

**Now, we respond in bold to your specific comments (in italic). Attached, you'll find the revised version of the manuscript with changes indicated in bold.**

*The paper emphasizes the fact that the triple-wavelength lidar observations of depolarization together with extinction-to-backscatter ratios of Saharan dust were done for the first time. However, for scientists as myself, who is not specialist in lidar instrumentation, it is not fully clear the importance of this new advanced technique. I would encourage authors to emphasize the achievements. For example, as a reader I have following questions:*

*What is the real advantage of the new technique? I could see triple-wavelength lidar observations of depolarization together with extinction-to-backscatter ratios of Saharan dust in other papers, for example, in Muller et al., (2012)?*
*Is there any important differences (comparing with previous studies, e.g. Muller et al., 2012) were revealed? If yes, then what are those differences? May be, the new technique mainly confirms the previous results? If so, this should be stated.*

**The extinction coefficient at 1064 nm could not be measured until the introduction of the rotational Raman method at this wavelength by Haarig et al., AMT 2016. The rotational Raman technique had been applied previously to other wavelengths such as 355 or 532 nm, e.g., by Veselovskii et al., AMT 2015. Former studies like Müller et al., 2012, used solely photometer observations to retrieve the lidar ratio at 1064 nm or combined photometer and lidar approaches like Tesche et al., Tellus B, 2009. Müller et al., 2012, clearly states "[107] We do not have measurements of the lidar ratio at 1064 nm" Unlike those retrieval results of fresh Saharan dust, our measurements in transported Saharan dust point to higher values of the lidar ratio at 1064 nm. In this way, they confirm the photometer retrievals (AERONET and GRASP) for the presented case study.**

*If I understood correctly, previously the measurements at 1064 were not simultaneous, but possible with few minutes of delay? If that is really a case, how critical is to have really simultaneous data, specifically thinking that more than 3 hours are required to*

*get some meaningful data is necessary.*

**Previously, there were no measurements of the extinction coefficient at 1064 nm. The combined lidar and photometer approaches of Tesche et al., 2009 and Mamouri and Ansmann, 2017, were the best estimates of the dust lidar ratio at 1064 nm until now. They clearly show the need of having that lidar ratio.**
**It is not so important to have simultaneous measurements, because 2-3 hours of averaging during nighttime are necessary to derive the extinction coefficient at 1064 nm.**

*Why in the figures 1 and 3 the different lidar characteristics are provided in different altitude ranges? These parameters can't be retrieved simultaneously even with this advance technique? If yes, this shows some remaining serious limitation of the measurements. Also, the spectral dependencies of depolarization ratios, lidar ratios and other properties are different and often contradicting to each other especially at low and high altitudes.*

**The different altitude ranges in Figure 1 and 3 reflect the fact that the overlap function affects the extinction, but not the backscatter.**
**The calculation of the extinction coefficient is based on the Raman signal only (387, 607 and 1058 nm). Therefore, it is affected by the overlap of the emitter and receiver field of the view of the lidar system. The overlap of the BERTHA is complete at 800-1000 m. Below this height, no extinction can be measured. Whereas the backscatter coefficient is calculated by using a signal ratio. In this way, the overlap function cancels out, because both signals are affected in the same way.**

**The spectral dependence of the PLDR and the lidar ratio reflects changes in the optical properties of the distinct aerosol layers. These vertical differences might be caused by different mixing ratios of dust and pollution, by different source regions or transport paths of the mineral dust.**

*In my opinion, there are several weaknesses in the methodology of comparisons of lidar data with AERONET in the present version of the manuscript. They need to be addressed and corrected:*

*First of all, the AERONET information content is significantly different from that of lidar: AERONET is measuring aerosol properties in total column while lidar in specific altitudes and doesn't observe not negligible aerosol layer near ground;*

**You are right. In the revised version we clearly state, that the fine-mode pollution found in the wintertime planetary boundary layer over Leipzig, affects the columnar observations and that we can not directly compare the layer-based lidar results with the columnar photometer results. We added the statement in lines 274-279:**
**"However, in the present case the explanation might be simpler. The sun photometer measures columnar values whereas the profile measurement of lidar permits to focus on the dust layer only. The fine-mode aerosol pollution of the wintertime boundary layer at Leipzig can significantly influence the AERONET lidar-ratio products, especially at the shorter wavelengths of 440 and 380 nm. Therefore, the columnar lidar ratio should increase towards the UV."**

*Lidar measures signal in back scattering, while AERONET radiances are measured mostly in forward hemisphere with maximum scattering angle of ~ 140-150 degrees. The properties provided by the AERONET retrieval at 180 degrees are not directly constrained by the measurements.*

**Yes, the 180° backscatter direction is not accessible for photometers and lidar systems are limited to this special angle. To mention this difference, we added a statement in the conclusion section, lines 347-352:**
**"In contrast to lidar systems which are only operating at 180° backscatter angle, this angle is not accessible for the photometer. Therefore, the photometer retrievals are not**
**optimized for this special angle. Joint inversions of the lidar and photometer data as done in Lopatin et al. (2021) would be desirable."**

*The above differences should be considered and discussed seriously in the AERONET and lidar comparisons:*

*AOD and AE from AERONET are nearly directly measured by photometers and very reliable. It would be good if integrated extinction values and their spectral dependence compared AERONET AOD and AE, before comparing more complex retrieval products.*

**We agree and added a in Section 4 "Discussion" a new subsection " Backscatter and extinction Ångström exponent", where we compare the AOD and AE for the 22 February case.**

*The reported aerosol lidar ratios and depolarizations are not fully direct measurements and rely on some caveats and assumptions in processing. It would be nice if those assumptions and their accuracy are discussed. For example, lidar community tends to consider desert dust properties spectrally independent while, in reality, it is not true. Even AE of extinction for dust is rarely zero, and it can be notably positive or negative.*

**In Figure 1 + 3, we added the vertical profiles of the extinction Ångström exponent (EAE) and backscatter Ångström exponent (BAE). Within the dust layer, the EAE is very close to zero in the 532/1064 range (see also Table 1). The EAE in the 355/532 is also very close to zero for the pure dust case (22 Feb) and around 0.33 for the polluted dust case (3 March). These results were obtained without further assumptions. However, the dust layer filled not the whole atmospheric column. For example, the EAE 355/532 in Fig. 1 shows an increase below the dust layer base at around 2 km, indicating a different aerosol situation below.**

*In the considered observations, the observed aerosol seems to be not a pure dust in the entire vertical column. How the analysis assure that lidar characteristics derived at specific altitudes are for the same dust as seen in total column by AERONET?*

**If we are not in a dessert, it cannot be assured that the whole atmospheric column is filled with mineral dust. In continental Europe we have to deal always with the boundary layer aerosol, if we want to compare columnar photometer observations with our lidar measurements. However, the dust event of 22-23 February was quite strong, so the dust layer should dominate the columnar optical properties. We added several statements concerning the fine-mode pollution in the boundary layer to explain differences in the observations, especially of the lidar ratio in the UV, see lines 274-278, 319-320, 354-355.**

*The authors write: "The drop (I ASSUME "increase") of the imaginary part of the refractive index at 440 nm compared to 675 nm is too strong in the AERONET inversion procedure, resulting in too high lidar ratios at 440 nm. In-situ studies could not confirm the spectral slope of the imaginary part used in AERONET inversions as discussed by Müller et al. (2012)." First, all AERONET "retrieves" imaginary part, not "uses" as the authors imply. Therefore, the spectral dependence is induced by the necessity of measurement fit. In these regards, I think the sensitivity to the imaginary part is likely higher in AERONET data than in lidar ones. Also, the obtained spectral slope tends to agree with the majority of other analyses of the dust (Shettle and Fenn 1979;*

*WMO1983, Patterson et al. 1977; Sokolik et al. 1993; Koepke et al. 1997; Sokolik and Toon 1999, and there are many more recent data). In contrast, as I mentioned above the lidar community tends to assume no spectral dependence for desert dust properties. In these regards, I would suggest the authors to elaborate this aspect and if there is full confidence in the results to put a clear statement suggesting to use neutral or less pronounced spectral dependence for imaginary part of the dust. Some more extensive literature review of this subject is also desirable.*

**The sentence was corrected and the second sentence about the in-situ studies of the imaginary part of the refractive index was removed. Overall, we realized that the fine-mode pollution included in the columnar AERONET observations prevent us from drawing firm conclusions from the lidar ratio comparison in the UV. For the longer wavelengths, the lidar observations confirm the spectral slope retrieved by AERONET (and GRASP). However, the small urban haze particles have a stronger effect on the shorter wavelengths. So, we don't argue about the imaginary part of the refractive index any more. We have to be in a desert, to have the whole column filled with mineral dust.**

**The lines 271-279 state now:**

**"The enhanced lidar ratios in the UV retrieved by AERONET were already discussed by Shin et al. (2018): The increase of the imaginary part of the refractive index at 440 nm compared to 675 nm is too strong in the AERONET inversion procedure, resulting in too high lidar ratios at 440 nm. However, in the present case the explanation might be simpler. The sun photometer measures columnar values whereas the profile measurement of lidar permits to focus on the dust layer only. The fine-mode aerosol pollution of the wintertime boundary layer at Leipzig can significantly influence the AERONET lidar-ratio products, especially at the shorter wavelengths of 440 and 380 nm. Therefore, the columnar lidar ratio should increase towards the UV."**

*One of the weakest points of the paper is comparison with so-called GRASP retrievals. First of all, I would like to emphasize that GRASP and AERONET retrievals have the same origin (Dubovik and King, 2000 and Dubovik et al. 2006). Indeed, Giles et al. (2019) was reporting only updates in quality assurance of the AERONET results but not the change of the retrieval, Sinuyk et al. (2021) outlined few new elements in the retrieval but emphasized that overall the retrieval methodology remains the same as in Dubovik and King, 2000 and Dubovik et al. (2006). Therefore, for the same data GRASP and AERONET retrievals should not show any significant differences. Certainly, GRASP is more flexible and can easily use the different (more complete observations than a 4 wavelengths) and this is why GRASP was used in the paper, I suppose… However, what was the RATIONAL for comparing lidar results with results of inversion of AOD data only (i.e. no angular sky radiances) as done by GRASP-AOD approach by Torres et al. (2017)? It is evident, that AOD HAS practically NO SENSITIVITY to aerosol properties at 180 degrees!!! Torres et al. (2017) clearly stated that GRASP-AOD strongly relies on the a priori assumptions about complex refractive index and particle sphericity. It is interesting that the authors see the better agreement of GRASP lidar ratios with the lidar data. This probably shows nice potential of GRASP-AOD retrieval once the constraints are correctly assumed by Torres et al. (2017). However, overall, I do not see much justifications of comparing AOD only retrievals with lidar results!*

**The authors were aware of common nature of GRASP and AERONET retrievals and that you have tried to show that in the paper (lines 114-115).**

**The so-called GRASP retrievals in our paper included AOD + radiance measurements. This information is now clearly stated in the new version of the**

manuscript and we have added other references to GRASP (Dubovik et al., 2014, 2021) in the paper since the use of the reference Torres et al. 2017 only can be confusing. The use of GRASP algorithm is justified since it allows us to include 380, 500 and 1640 nm measurements (both radiances and AOD) and it is an open code with similar approach of AERONET.

Since we include radiance measurements the refractive index and the sphericity are not pre-set and are retrieved following the same strategy as AERONET but with the extra information provided by the use of more wavelengths.

In the instrument section, we rephrased the following paragraph (lines 112-117):
"Besides the standard AERONET data analysis procedure (Dubovik and King, 2000; Dubovik et al., 2006; Sinyuk et al., 2020), the related GRASP retrieval scheme was applied (Dubovik et al., 2014; Torres et al., 2017; Toledano et al., 2019; Dubovik et al., 2021). GRASP is based on a similar approach as the standard 115 AERONET retrieval (Dubovik and King, 2000), but using optical information (AOD and radiances) at the wavelengths of 380, 500 and 1640 nm additionally. From both algorithms the column lidar ratios and depolarization ratios at several wavelengths were retrieved."

*From my viewpoint, the comparisons of GRASP results with the lidar ones would be much more convincing if GRASP would be used for inverting AEONET AOD and radiances at more wavelengths (even if at some new channels only AOD data would be added). Moreover, I would think that the best approach would be to use GRASP for simultaneous inversion of the lidar data and AERONET dada as illustrated by Lopatin et al. (2013) and especially Lopatin et al. (2021). Such approach would be the most appropriate to reveal the strength and shortcoming of GRASP aerosol model, since would show if the model can adequately reproduce all (lidar and photometer) data or not.*

We agree, that joint inversions of lidar and photometer data using GRASP would be the way forward to inspect the GRASP model assumptions. We are looking forward to a collaboration with Anton Lopatin about this task.
In the conclusion (lines 351-352), we paved the way forward to future collaboration:
"Joint inversions of the lidar and photometer data as done in Lopatin et al. (2021) would
be desirable."

*The authors emphasize serious limitations of Dubovik et al. (2006) spheroid model to reproduce the lidar measurements. Based on my personal experience I would probably agree that Dubovik et al. (2006) model would have difficulties to reproduce nonmonotonic spectral dependence of depolarization ratio, as well as, the highest values of the depolarization ratio. However, I do not think that the comparisons of lidar data with AERONET and GRASP shown in the paper can be considered as a solid justification for such a statement. Once again, I would suggest trying the joint inversion of lidar and photometer data by GRASP as demonstrated by Lopatin et al. (2013, 2021). By the way, Lopatin et al. (2021) shown that using direct lidar measurements, e.g. volume depolarization instead of aerosol depolarization ratio results in better comparisons between photometric and lidar results.*

The lidar ratios are well reproduced. In the case of the particle linear depolarization ratio (PLDR), it is more difficult, because of the strong effect of the non-sphericity of mineral dust particles. Especially the spectral slope of the PLDR of dust as reported from several studies (Fig. 5a), seems to be difficult to reproduce with the model assumptions used in the AERONET retrieval. The use

**of more wavelengths seems to improve the spectral slope as shown by Toledano et al., 2019. We have to admit, that the PLDR at 1064 nm was very low in the present case (see as well Fig. 5a), now stated in lines 311-312.**

*Minor comment:*

[revised manuscript text omitted]